# Machine Learning for Future Subtyping of the Tumor Microenvironment of Gastro-Esophageal Adenocarcinomas

**DOI:** 10.3390/cancers13194919

**Published:** 2021-09-30

**Authors:** Sebastian Klein, Dan G. Duda

**Affiliations:** 1Gerhard-Domagk-Institute for Pathology, University Hospital Münster, 48149 Münster, Germany; 2Institute for Pathology, Faculty of Medicine, University Hospital Cologne, University of Cologne, 50931 Cologne, Germany; 3Edwin L. Steele Laboratories for Tumor Biology, Department of Radiation Oncology, Massachusetts General Hospital, Harvard Medical School, Boston, MA 02478, USA

**Keywords:** gastric cancer, esophageal cancer, gastro-esophageal, machine learning, tumor microenvironment, deep learning, artificial intelligence, immunotherapy, omics

## Abstract

**Simple Summary:**

We summarize the main components of the tumor microenvironment in gastro-esophageal adenocarcinomas (GEA). In addition, we highlight past and present applications of machine learning in GEA to propose ways to facilitate its clinical use in the future.

**Abstract:**

Tumor progression involves an intricate interplay between malignant cells and their surrounding tumor microenvironment (TME) at specific sites. The TME is dynamic and is composed of stromal, parenchymal, and immune cells, which mediate cancer progression and therapy resistance. Evidence from preclinical and clinical studies revealed that TME targeting and reprogramming can be a promising approach to achieve anti-tumor effects in several cancers, including in GEA. Thus, it is of great interest to use modern technology to understand the relevant components of programming the TME. Here, we discuss the approach of machine learning, which recently gained increasing interest recently because of its ability to measure tumor parameters at the cellular level, reveal global features of relevance, and generate prognostic models. In this review, we discuss the relevant stromal composition of the TME in GEAs and discuss how they could be integrated. We also review the current progress in the application of machine learning in different medical disciplines that are relevant for the management and study of GEA.

## 1. Gastro-Esophageal Adenocarcinoma (GEA)—An Introduction

### 1.1. Tumor Microenvironment (TME)

Tumors may be seen as an abnormal organ, forming as a result of close interaction of cancer cells with the surrounding tissue [1]. Cancer initiation is a process primarily driven by genetic alterations of somatic cells at the site of tumor origin, but concomitant responses occur at the cellular level altering the TME [2,3]. For instance, increased expression of pro-inflammatory cytokines such as interleukin (IL)-1, IL-6, interferon (IFN)-γ, and tumor necrosis factor (TNF)-α, leads to recruitment and activation of several cell types of immune and stromal cells that promote adaption of residual cells [4,5,6]. As tumors grow, genetic alterations may increase in complexity [7]. In parallel, imbalances in nutrient supply (hypoxia), as well as acute-to-chronic inflammation reveal a dynamic shaping of this process [8,9]. Therefore, using snapshot information on the TME appears unlikely to be useful, as the TME is dynamically altered by multiple factors (Figure 1). These complex factors include nutrient supply, genetic alterations, and cytokine/chemokine gradients, all of which can show temporal and spatial intratumoral heterogeneity [10,11]. As current cancer treatment paradigms are shifting from “one-size-fits-all” therapeutic strategies to approaches based on precision medicine, it is of crucial importance to gain a relevant understanding of the TME to suggest the best therapy for every single patient at an individual level and specific stage of progression, as well as to discover novel therapies in the near future.

The TME of established solid tumors consists of different cellular components, including activated fibroblasts [32,33], immune cells (lymphocytes, macrophages, dendritic cells), and endothelial cells, all with distinct functions, reviewed elsewhere [34,35,36,37]. Other mechanisms in early tumorigenesis and late disease progression include epithelial-to-mesenchymal transition (or EMT), a process whereby tumor cells may undergo specific phenotypical changes, which is reviewed elsewhere [12,13].

The therapeutic promise brought by new immunotherapies, in particular those targeting immune checkpoint molecules or administration of antitumoral immune cells, has shifted the interest of TME studies towards a better understanding of the immune TME [13]. Here, the abovementioned cellular components can show distinct phenotypes and anti- or pro-tumor activities, in a cancer type- and site-dependent manner [14].

We will briefly discuss methodical approaches using machine learning (ML) that may allow to provide a more integrative view on the TME and facilitate this deeper understanding. We performed a focused publication search using PubMed database by using keywords such as tumor microenvironment, gastric cancer, esophageal cancer in addition to a dedicated search for literature on the topic of machine learning using the following search query: (gastric cancer OR esophageal cancer OR gastro-esophageal cancer OR gastro-esophageal junction cancer OR esophageal adenocarcinoma) AND (machine learning OR artificial intelligence OR deep learning).

### 1.2. Brief Overview of GEA

#### 1.2.1. Introduction

Worldwide, the incidence of gastric cancer ranks 6th and the number of related deaths ranks 3rd; esophageal cancer ranks 10th and 6th, respectively, according to the global cancer statistics of 2020 [15]. Pathologically, adenocarcinomas of the lower esophageal tract and gastric adenocarcinomas can have diverse etiologies. Adenocarcinomas of the esophagus are primarily thought to develop as a result of gastro-esophageal reflux disease [16,17,18], while adenocarcinomas of the stomach may arise in association to infection with Epstein–Barr virus or helicobacter pylori or may develop in a setting of cancer predisposition syndrome—which is discussed elsewhere [17,19,20]. However, there is often a mutational overlap between gastric and esophageal adenocarcinomas [21]. Molecular classifications include esophageal adenocarcinomas into the chromosomally unstable subtype of GEAs [22,23]. Clinically, the standard therapy for GEA includes perioperative therapy, surgery, and radio-chemotherapy depending on tumor stage and patient characteristics with additional potential novel therapies which are discussed in the following section [24].

#### 1.2.2. Current and Future Therapeutic Concepts in GEA

Despite targeting oncogenic alterations of tumor cells in GEA, progress has been limited in unselected patient populations [21,23,25]. Targeting the TME has become of great clinical interest, fueled by the development of effective anti-angiogenic therapies and immunotherapies [26,27,28,29,30,31].

Although initial trials with the anti-vascular endothelial growth factor (VEGF) antibody bevacizumab initially provided negative results, the anti-VEGF receptor (VEGFR)-2 blocking antibody ramucirumab demonstrated efficacy in GEA alone and with chemotherapy [38,39,40]. Interestingly, in a recent meta-analysis, anti-angiogenic therapies added a benefit to overall survival in these cancers. It is also worth noting the recent positive results with the use of the multitargeted tyrosine kinase inhibitor (mTKI) apatinib, an agent with anti-VEGFR activity, in Chinese GEA patients [41,42].

More recent efforts in oncology have centered on the development of therapies that use immune-checkpoint inhibitors (ICI) [26,28,43]. A key mechanism of action of ICIs is to alleviate immune-cell exhaustion that leads to immunogenic tolerance towards tumor cells. Blocking these molecules would enhance anti-tumor immunity, and indeed, this concept is supported by recent successes in GEA patients [44,45,46,47]. This has led to the approval of ICI in advanced stage GEA cancers expressing PD-L1 [48,49,50]

A particularly promising therapeutic avenue appears to be the combination of ICIs with anti-angiogenic agents via normalization of the vasculature and reprogramming of the immune TME, reviewed elsewhere [51,52,53,54,55,56]. Interestingly, this concept is now being evaluated in several clinical trials in GEA with promising initial results [51,57].

#### 1.2.3. The TME of GEA

Chronic inflammation can be seen as a major risk factor for developing GEA [58,59,60,61]. Here, cytokines released both locally and systemically (for example in patients with underlying conditions such as obesity) create a disbalance of cellular stress [62,63,64,65]. In the process of tumor initiation, several cytokines shape a pro-tumorigenic TME with accumulation of myeloid-derived suppressor cells (MDSC) in the very initial phase, in addition to macrophages, reviewed elsewhere [36,66,67,68,69,70,71]. For instance, the pro-fibrotic and immunosuppressive transforming growth factor (TGF)-β, or the pleiotropic immune mediators IL-1 and IL-6, mediate this process [5,6,72,73,74,75].

With more men than women being affected by GEA (6.6 [8.2]/1.8 [3.8], region-specific incidence for age-standardized rates by sex for esophageal cancer [stomach] in 2020, Western Europe [15]), one should also appreciate that sex may be considered as a variable in future trials and clinical management of GEA patients (Figure 1) [76,77,78,79,80]. Given the relevance of sex differences in cancer mortality, molecular and genetically, as well as pharmaceutically, future studies need to define the underlying mechanisms for these differences when studying the TME of GEA [81].

#### 1.2.4. Biomarkers in GEA

As precision oncology emerges, classification of GEA such as the TCGA—chromosomal instable subtype (50%), microsatellite instable (MSI) high subtype (22%), genomically stable subtype (20%), Epstein–Barr virus-positive subtype (9%) among other classifications—are increasingly being used for prognostication [21,82,83,84,85,86]. Unfortunately, these classifications have not yet been fully translated into improved therapeutic regimens, although MSI high subtypes and Epstein–Barr virus-positive cases show increased rates of response to ICI [21,82,83,84,85,86]. Additional genes that may help GEA patient stratification for targeted therapies include *EGFR* and Her2/neu (*ERBB2*) [87,88,89,90,91,92,93]. Finally, tissue biopsies, taken at initial diagnosis, may also be used to identify expression signatures to predict response primarily to neoadjuvant therapy [94,95,96,97,98].

Given the increasing role of immunotherapy in the treatment of GEA, inflammatory phenotypes and biomarkers that are linked to pro/anti-tumoral properties are investigated in current studies [99,100]. Mechanistically, acquired, or intrinsic resistance to immunotherapy is a complex process. Thus, a generic biomarker to predict response to ICI remains elusive. However, PD-L1 combined positive scoring (CPS), appears to identify patients with GEA that may respond to anti-PD/L1 antibody immunotherapy. In addition, there is interest in defining the role of the number of somatic mutations (tumor mutational burden) as a biomarker for ICI [48,101,102]. Many prediction models of response to ICI consider the frequency of tumor-infiltrating lymphocytes (TILs) [4,103,104,105,106]. Moreover, integrative diagnostic approaches that combine several omics techniques have been shown to increase prediction to response to ICI therapy, including tumor-mutational burden (TMB) or neoantigen burden, to identify tumors with pro-immunogenic properties [102,103,104,107,108,109,110,111].

## 2. Machine Learning—Basic Concepts, Specific Applications, and Future Directions in GEA

### 2.1. Basic Concepts of ML

ML has gained recent interest within medical research, as large (annotated) datasets have become available, hardware components have allowed more complex models to be trained and a broad distribution and accessibility of code and examples have emerged and allowed the field to grow rapidly. Within the following section, we will first introduce the basic concepts of ML and then briefly review their application in GEA.

#### 2.1.1. Supervised Learning

The term “supervised” refers to the technique where a model is supplied with data (known as features; for instance, genes with quantile normalized array data) and a target variable is defined (outcome; for instance, response to therapy). Depending on the design of the algorithm and the nature of the type of the target, the algorithm may return a class (responder/non-responder) or a continuous variable (time to relapse/score). The application of supervised learning with certain deep convolutional neural networks is occasionally referred to as artificial intelligence [112].

#### 2.1.2. Unsupervised Learning

Here, no target variables are defined that a given algorithm is trained on. Rather than proposing classes or scores, unsupervised learning methods are primarily used to show (visualize) differences and similarities between samples. Commonly, unsupervised learning is used to reduce the complexity of a dataset for subsequent supervised learning (feature selection). However, within data exploration, unsupervised learning may be applied to study relationships and understand connections that need to be uncovered in each dataset, including gene network analyses [113,114]. For instance, visualization techniques using principal component analysis (PCA) and *t*-statistic stochastic neighbor embedding (t-SNE) are widely applied in the biomedical field, especially given the growing interest in single-cell RNA/DNA sequencing [115,116,117,118].

#### 2.1.3. Choosing the Right Approach for the Right Kind of Datatype

The application of different DL models for supervised learning has allowed major advances to within the biomedical field (Figure 2). Especially for object detection, classification, and (semantic) image segmentation DL allowed major progress to be made. Although DL shows advantages to solve problems of unstructured data, classical regression and classification models are still useful. Linear regression models have the advantage to allow revealing the contribution of variables. This can be of interest (in the medical field) to potentially allow quality control of the variables of interest or to even collect the given variables actively for future studies.

### 2.2. Specific Application of ML in GEA

So far, several studies have applied regression models and DL models to address different kinds of medical problems in GEA. We have divided the different disciplines and diagnostic modalities to show the potential application of ML for GEA. A summary of the disciplines and the type of problems addressed by them are summarized in Figure 3.

#### 2.2.1. Epidemiology, Radiation Oncology, and Blood Biomarkers

Yoon et. al., used a logistic regression model and a supportive vector machine to predict excessive muscle loss during neoadjuvant radio-chemotherapy by analyzing patients’ blood samples and body mass index [119]. Interestingly, ML may also be used to propose risk factors for anastomotic leakage after esophagectomy [120]. Other attempts included using DL to identify optimal dosing of radiotherapy in GEA or defining the optimal target volume and organs at risk [121,122,123,124,125]. A dedicated analysis by Rahman et. al., used a random survival forest model by utilizing a dataset of more than 6000 patients to identify long-term survivors after esophagectomy [126]. Aslam et al., applied an autoencoder to a breath analysis and showed that this approach may be used for early GEA detection [127].

By applying an Extreme Gradient Boosting (XGBoost) technique, Leung et al., predicted the risk of GEA development after *Helicobacter pylori* eradication [128,129]. Non-invasive techniques can also be used, in combination with a gradient-boosting decision tree, to build a predictive model identifying patients with GEA [130]. Other studies proposed an ML-based approach to identify patients who would require early readmission after surgical intervention of GEA [131].

#### 2.2.2. Endoscopy-Based Approaches

Several studies trained CNNs to aid early detection in GEA, recently summarized in a meta-analysis that found superiority of applying DL for detection of Barrett’s esophagus [132,133,134,135,136,137]. Currently, clinical trials are already investigating its sensitivity and specificity, if applied in a clinical setting, with several studies showing DL models to identify early GEA [138,139]. In detail, 3D endoscopy imaging techniques, in combination with DL, may be applied to quantify the depth of Barrett’s esophagus [140]. Of clinical relevance, another study found spectral endoscopy combined with DL more sensitive and specific to detect dysplastic vs. non-dysplastic Barrett than previous techniques [141]. Similarly, DL models have been used for the detection of intramucosal GEA using [142].

To identify individual patients and potential risk factors for recurrence of GEA after surgical intervention, Zhou et al., applied several regression/classification models and identified clinical variables that are associated with an increased risk [143]. A recent meta-analysis of several studies in Asian populations found that endoscopic imaging may also be analyzed to detect the presence of *Helicobacter pylori* infection [144].

#### 2.2.3. Genomic-Based Approaches

Several studies used gene expression signatures, with array techniques or using NanoString, in addition to other techniques of RNA sequencing, to identify patients who would respond to chemotherapy in GEA [145]. In parallel, Chen et al., proposed seven immune-related genes to predict prognosis in GEA by applying a regression analysis [146]. Moreover, supportive vector machines have been applied to identify novel markers from circulating tumor cell-free DNA [147]. Recently, a multi-omics approach could identify responses to neoadjuvant therapy in GEA [148].

By following a complex combination of (similarity) clustering, Yuan et al., identified previously unrecognized non-coding long RNAs (lncRNAs) in gastric cancers [149]. In parallel, Li et al., compared several classification/regression models to identify novel lncRNAs in GEA [150]. Usually, a combination of both supervised and unsupervised techniques is used to identify subgroups of patients. For instance, to detect immunological subtypes of gastric cancer Chen et al., used a K-means clustering algorithm to detect subgroups based on RNA expression data and then trained a CNN to detect these subtypes using virtual-whole-slide images [120].

Genome-wide association studies identified novel susceptibility genes to gastric cancer using a random forest model [151]. By using several clustering algorithms of different sources of genomic data of 70 gastric cancer patients, Wang et al., proposed the detection of molecular subtypes in GEA [152]. After combing gene expression data and DNA methylation data for subsequent feature selection, Zhang et al., trained a model to detect novel biomarkers for discriminating between tumor and normal mucosa [153].

Owen et al., harvested mucosa tissue from different anatomical locations of the stomach to identify an overlap between Barrett mucosa and found an association to submucosal glands by single-cell RNA sequencing [154]. Here, and within other studies applying single-cell RNA sequencing, SC3 consensus clustering has been applied as an unsupervised learning method to allow the identification of certain genes that could distinguish common alterations in mucosa tissue [154,155].

#### 2.2.4. Radiology-Based Approaches

Another example of ML applications in GEA is radiology, where different imaging modalities are used, most frequently CT imaging. For instance, CT imaging objects have been used to predict response to neoadjuvant therapy or to characterize tumor stromal components [156,157]. In a recent study, Lin et al., trained a CNN to detect lymph node metastasis by analyzing perioperative CT images of patients with gastric cancer. In addition, and relevant to potential therapeutic de-escalation therapy and patient surveillance, CT scans may also be used to monitor responses to (neoadjuvant) chemotherapy in GEA [158,159]. Other attempts involved training a model to aid the detection of GEA using CT scans [160].

Liu et al., followed an integrative approach of combining preoperative biomarkers including tissue biopsies, tumor markers, and CT image objects to predict lymph node metastasis in GEA by applying regression analysis and combined this to a multivariate model [161]. In parallel, similar image object information have been used to predict the risk of peritoneal metastases using gradient boosting machines [162]. Others applied DL models to detect metastasis using CT image objects, in addition to adequate staging [163,164].

#### 2.2.5. Digital Pathology and Virtual Microscopy-Based Approaches

Current examples that facilitate virtual whole slide images from regular H&E stains, that are generated within routine pathology workflow, include subtyping gastric cancer by convolutional neural networks [165,166]. In a recent study, Wang et al., trained a DL segmentation model to identify tumor regions within lymph nodes of gastric cancer patients and showed that this may serve as an interpretable independent prognostic factor in GEA [167]. In a study from our group, we developed a decision support system that combines morphological image operations to detect areas of relevance in large virtual whole-slide-image objects and proposes areas of *Helicobacter pylori* presence that can increase the sensitivity of identifying HP in gastric cancer biopsies, both of standard H&E staining and specialized Giemsa staining [168].

Park et al., trained a DL algorithm to identify gastric cancers in endoscopy biopsy specimens and showed that the system can increase time to diagnosis and may be potentially applied in countries with a lack of specialized pathologists [169]. Similarly, a recent multicentric study built a DL-based algorithm to aid in the diagnosis of gastric cancer and applied this using data from different scanners and different hospitals showing its generalization [170]. Sali et al., compared supervised and unsupervised DL-based models to identify dysplastic and non-dysplastic Barrett’s Esophagus by analyzing virtual whole-slide images and found that unsupervised models achieved better results in comparison to supervised DL [171]. To reveal potential prognostic biomarkers of the TME in GEA, Meier et al., applied a DL model using tissue microarrays of a Japanese cohort [172]. Recent articles summarized potential requirements to more widely applying DL in gastrointestinal pathology [173,174], in addition to a systematic review highlighting applications of virtual whole-slide image analysis in GEA [175].

A retrospective multicentric study by Muti et al., built and validated a DL model predicting microsatellite and Epstein–Barr virus-associated GEA subtypes within a cohort of more than 2500 patients using scanned H&E whole-slide-images [176]. Although these advances appear to align with a recent success story of applying DL on histological images to predict microsatellite instability within colorectal cancer, published by the same group, it remains to be determined whether a molecular classification using DL would add benefit to the treatment of patients within prospective multicentric trials [177]. However, these proof-of-concept studies clearly indicate that molecular phenotyping using histological images may be of clinical interest. Future trials need to determine the exact value of these techniques as screening or eventually as additional parameters.

Broad and deep genetic sequencing efforts of tumor tissues and additional molecular analyses by “histological genotyping” could provide biomarkers of response but identify new targets for the treatment of a given patient. However, is it possible, for instance, that patients classified as microsatellite stable with help of DL applied on histological images show resistance to ICI despite the suggestion of response by genetic-based classification? Could we identify patients more likely to respond to ICI by building DL models end-to-end for treatment outcome as an alternative to more cumbersome molecular subtyping?

### 2.3. Current Status, Future Directions and Challanges of ML in GEA

#### 2.3.1. Current Status of Machine Learning

Different ML models and techniques have been applied by several medical disciplines for object detection, segmentation, classification, and prognostic modeling using structured and unstructured data (Figure 2 and Figure 3). Many of these applications appear to operate parallel to biomarkers that are already established. For instance, while the detection of molecular alterations using imaging objects may reduce costs and can potentially save time, there are already established techniques with approved drugs and sufficient sensitivity to allow identification of patients that will qualify for targeted therapies [92].

However, given the complexity of human cancers and their evolution during progression, where therapeutic pressure can select for drug-resistant clones and alter the TME, it would be of great relevance to use ML models to provide a more integrative view of the TME and its dynamic changes (Figure 1) [178,179,180].

#### 2.3.2. Challenges and Future Directions

In addition to collecting tumor biopsies from cancer patients during treatment, an approach requiring invasive surgical procedures that may not always be feasible, radiological imaging data, and blood biomarkers may be used to gain more information on the TME of GEA from different time points during disease progression and therapy [181,182,183]. This would initially require finding surrogate markers or to generate models that find correlates of different TME phenotypes from these (preferable non-invasive) measures. For instance, circulating T-cell might be used to understand mechanisms of immune evasion, in addition to cytokine in blood circulation [184]. We and others have studied angiogenic biomarkers that may be useful biomarkers for tumor vascularization and vascular function, to facilitate the use of immunotherapy and anti-angiogenic therapies alone or in combination [56,185,186].

In the future, combined omics approaches that integrate most data resources will be gathered to retrieve biomarkers using ML models that will add a more holistic view on cancer progression, treatment resistance, and therefore optimal therapeutic decision making. Despite single diagnostic modalities that are revealing diagnostic or therapeutic proof of concept, this will unleash the full potential of well-annotated and well characterized datasets providing clinicians with the necessary information for decision making during the management of GEAs (Figure 4).

Here, modern technologies may be used to decipher aspects of the TME providing information that are not accessible by other technologies. In detail, in a first discovery phase (Figure 4), application of several omics technologies in parallel will be necessary to identify molecular and spatial characteristics of GEA that help to identify subtypes of the TME aiding dedicated therapeutic approaches. While ML models will be used during this phase to discover these characteristics, unsupervised algorithms that allow the detection of relevant biomarkers will be necessary to guide diagnostic modalities in the application phase.

Although ML, and in particular the application of DL for unstructured imaging data, could increase the sensitivity and specificity of cancer diagnosis, with detection of molecular subtypes, identification of subgroups, and stratification of patients, these advances need to be validated and certified by regulatory bodies [187]. While many of the mentioned studies supplied proof of concept, a combination of biomarkers would require dedicated development for companion diagnostic applications.

At the same time, some of the mentioned models act as a “black box” where it appears to be difficult to decide what a given model detects, although reverse engineering, and study design, may allow for understanding the decisions of a network [188,189,190]. Interestingly, one may argue that increasing accuracy may be more important than understanding every aspect of a given model, especially given intra- and interobserver variability in medical decision making, in line with other examples of uncertainty, including mechanisms of drug action and disease mechanisms in medicine [191]. Other challenges that need to be overcome [192] include generating, validating, and applying models that can account for missing data points [193], highlighting the need for adequate preprocessing of data, in addition to normalization methods accounting for data variation [194].

To this end, medical disciplines need to be trained properly to understand the limitations of ML models. Likely, with the development of different DL architectures and ML techniques these applications need to undergo constant changes and adaption, requiring trained personal to address these challenges.

#### 2.3.3. Summary

In summary, the particularities of the TME of GEA need to be defined in a dynamic fashion to aid the current applications of ML in this cancer entity. This will only be possible if data from different disciplines are combined, aiming to gather relevant information that may inform therapeutic decisions. Here, we summarized the current understanding in different medical specialties and discussed the challenges that need to be overcome to provide a more integrative view of the TME of GEA and facilitate clinical translation for the improvement of personalized therapies in this aggressive malignancy.

## 3. Conclusions

Clearly, the ML field is still in its infancy and is focused primarily on discovery and proof of concept studies, but there is promise that the translation phase of ML is within the immediate future.

## Figures and Tables

**Figure 1 cancers-13-04919-f001:**
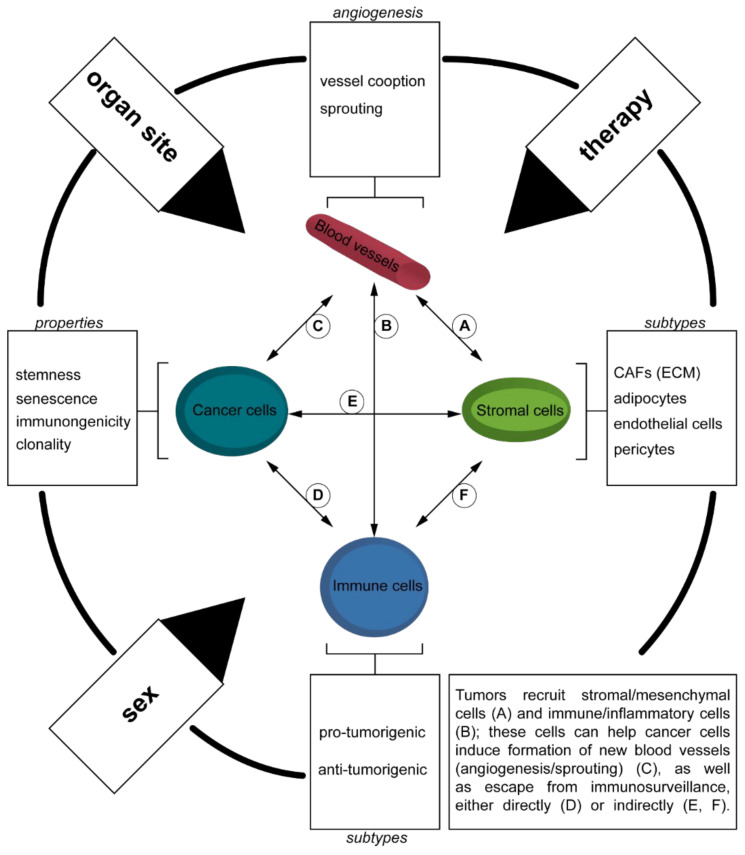
Integrative view of the tumor microenvironment. During cancer development, somatic mutations are acquired at the DNA level leading to uncontrolled cell growth. In detail, tumors are formed with clonal heterogeneity and potential stem-cell-like properties, in line with immune cell exclusive properties [9,12,13]. Within the context of the regional parenchyma (site of origin; localization), cytokines are released and promote new vessel formation, a process that may involve both sprouting angiogenesis as well as the co-option of preexisting vasculature, among other mechanisms [14,15,16]. The newly developed vasculature is usually immature and presents abnormalities, including increased permeability and poor perfusion (due to lack of pericyte coverage or due to collapse because of surrounding physical stress, which is deposited by specialized cancer-associated fibroblasts or CAFs) [6,17]. Vascular function is also influenced by the excessive deposition of extracellular matrix (ECM) components which may lead to blood vessel compression, altering oxygen supply, and decreasing therapeutic delivery and efficiency [18]. Hypoxia may increase genomic stress in cancer cells, in addition to other characteristics of cancer progression and therapy resistance [7,19]. In addition, the abnormal characteristics of blood vessels will attract inflammatory cells [20,21]. Moreover, cytokines and chemokine expressed by the cancer cells may attract immune cells, including lymphocytes, granulocytes, and macrophages, shaping a pro-tumorigenic TME, a process that may be influenced by sex in GEA [22,23,24,25,26,27] and administration of (cytotoxic) therapies [28,29], underlining a connective network between tumor cells, stromal cells, immune cells, and blood vessels. Together, as tumor site may change (as the progressing tumors metastasize at distant sites) and therapies increase selection pressure, the TME may undergo dynamic changes [30]. Moreover, administration of (cytotoxic) therapies potentially selects for cancer cell traits leading to senescence, a potential mediator of disease relapse [31].

**Figure 2 cancers-13-04919-f002:**
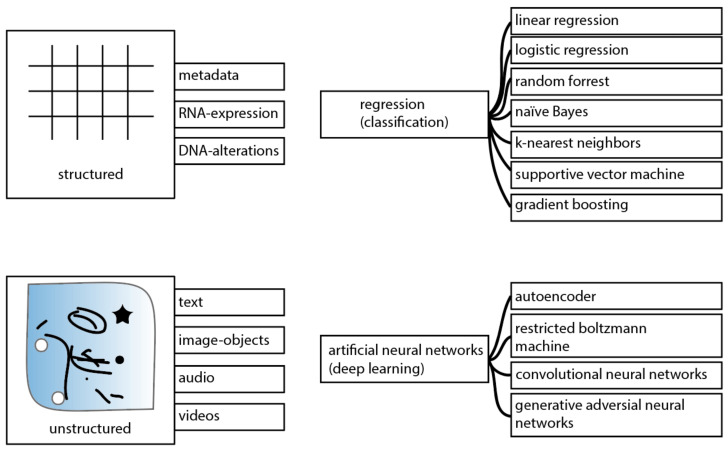
Overview of ML techniques that can be applied using an unsupervised learning approach. Regularly, tabular data (structured data), including genomics data are analyzed using regression or classification models. Notably, also structured data can be analyzed using deep learning (DL). As for unstructured data, where complexity increases, DL models are used in favor of regression/classification models. In particular, the field of computer vision and image analysis has shifted greatly to DL.

**Figure 3 cancers-13-04919-f003:**
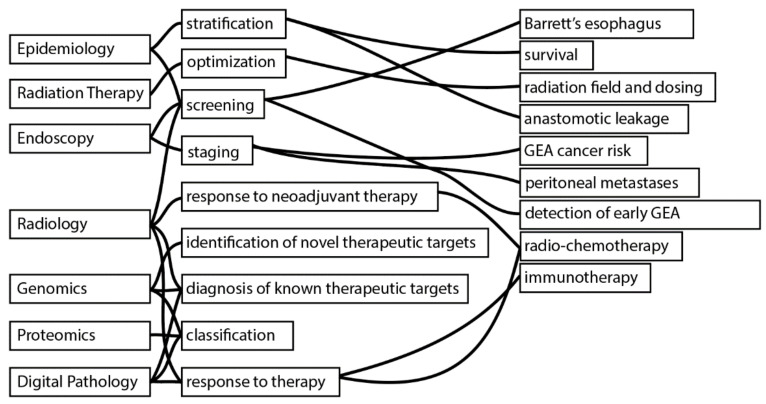
ML applications according to medical disciplines and diagnostic modalities in GEA. A description of studies following the given examples can be found in Section 2.2.1, Section 2.2.2, Section 2.2.3, Section 2.2.4 and Section 2.2.5. In summary, medical disciplines and diagnostic modalities including Epidemiology, Radiation Oncology (Therapy), Endoscopy, Radiology, Genomics, Proteomics, and Digital Pathology have shown how ML can be used to stratify patients for survival and complications of surgical intervention; optimization for dosing and radiation fields; screening for Barrett’s esophagus (dysplastic/non-dysplastic); early GEA detection; staging of cancer (peritoneal metastases, lymph node metastases); response to (neoadjuvant) therapy (radio-chemotherapy/immunotherapy) and discovery/diagnosis of novel/current therapeutic targets.

**Figure 4 cancers-13-04919-f004:**
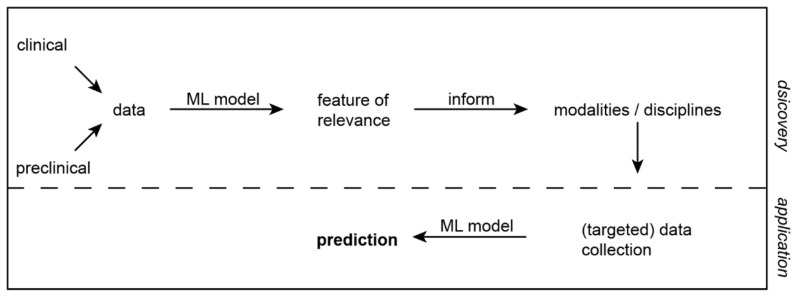
Future directions of machine learning in GEA. Within the discovery phase of machine learning, multi-omics approaches that are collecting data of various sources, including RNA and DNA sequencing, epigenetics, proteomics, imaging data, and metabolics will help to understand the TME, especially considering dynamic changes of cancer progression and treatment. These data will be integrated using machine learning and relevant information from all disciplines/modalities will be used to then inform the specialties what information (features) are necessary to apply specialized machine learning models that will predict individual disease traits, such as response to therapy and individual treatment strategies.

## Data Availability

Not applicable.

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
