# Peer review of "Machine Learning for Future Subtyping of the Tumor Microenvironment of Gastro-Esophageal Adenocarcinomas"

_cancers, 2021, doi:10.3390/cancers13194919_

Round 1

Reviewer 1 Report

In this review, authors attempted to summarize the microenvironment of GEA, and introduce the implication of machine learning in this aspect. There are some suggestions regarding the tumor microenvironment to further improve the manuscript:

1) Stromal cells should be more than just CAFs. Additional cell types should be listed in the Figure 1 to avoid misleading.

2) Transition of cancer cells is important for tumorigenesis, e.g. EMT, EndoMT, etc, but cannot find in the manuscript.

3) Macrophages largely contribute to the TME-driven cancer progression [PMID: 30692665], it should be included in section 1.2.3.

4) Neo-antigen is a way to identify new biomarkers for cancers, should discuss in 1.2.4.

5) Classic pathogenic signaling (e.g. TGF-beta, NF-kB; PMID: 34299192; 33114183) are important for promoting cancer via the TME, it should be discussed as a section 1.2.5.

6) Unsupervised gene networking analysis help to identify new therapeutic target PMID: 32788346; 29042082), it can be discussed in 2.1.2.

7) Single-cell RNA-sequencing is the latest approach to dissect the TME. Any implication of machine learning can be involved? Can be discussed in 2.3.2

Author Response

Point by point response to reviewers’ comments (in italics)

In this review, authors attempted to summarize the microenvironment of GEA, and introduce the implication of machine learning in this aspect. There are some suggestions regarding the tumor microenvironment to further improve the manuscript:

We thank the reviewer for the evaluation and useful suggestions.

1) Stromal cells should be more than just CAFs. Additional cell types should be listed in the Figure 1 to avoid misleading.

Thank you for pointing out his detail, we have depicted adipocytes, endothelial cells and pericytes in addition to CAFs in Figure 1.

2) Transition of cancer cells is important for tumorigenesis, e.g. EMT, EndoMT, etc, but cannot find in the manuscript.

We agree and added the following statement: “Other mechanisms in early tumorigenesis and late disease progression include epithelial-to-mesenchymal transition (or EMT), a process whereby tumor cells may undergo specific phenotypical changes, which is reviewed elsewhere [18, 19].” (lines 70-73)

3) Macrophages largely contribute to the TME-driven cancer progression [PMID: 30692665], it should be included in section 1.2.3.

We agree. We added a paragraph on immune cell subtypes in lines 63-66 “The TME consists of different cellular components, including activated fibroblasts [10,11], immune cells (lymphocytes, macrophages, dendritic cells), and endothelial cells, all with distinct functions, reviewed elsewhere [12-15]”.

We certainly agree with the reviewer that macrophages play an important role in cancer progression and therefore referenced two reviews highlighting the most important recent work on this topic: ”In the process of tumor initiation, several cytokines shape a pro-tumorigenic TME with accumulation of myeloid-derived suppressor cells (MDSC) in the very initial phase, in addition to macrophages, reviewed elsewhere [16,66-71]. “(see lines 127-130).

We also added the important references suggested by the reviewer.

4) Neo-antigen is a way to identify new biomarkers for cancers, should discuss in 1.2.4.

We agree with the reviewer that neoantigen burden - and its surrogate markers are - important feature biomarkers.  We include now the following statement: “Moreover, integrative diagnostic approaches that combine several omics techniques have been shown to increase prediction to response to ICI therapy, including tumor-mutational burden or even neoantigen burden, to identify tumors with pro-immunogenic properties [103-105,108-112].” (lines 162,163).

5) Classic pathogenic signaling (e.g. TGF-beta, NF-kB; PMID: 34299192; 33114183) are important for promoting cancer via the TME, it should be discussed as a section 1.2.5.

We agree with the reviewer and have added information on the current section of chronic inflammation: “Chronic inflammation can be seen as a major risk factor for developing GEA [56-59]. Here, cytokines released both locally and systemically (for example in patients with underlying conditions such as obesity) create a disbalance of cellular stress [60-63]. In the process of tumor initiation, several cytokines shape a pro-tumorigenic TME with accumulation of myeloid-derived suppressor cells (MDSC) in the very initial phase, in addition to macrophages, reviewed elsewhere [14,64-68]. For instance, pro-fibrotic and anti-tumorigenic TGF-beta, in addition to pro-angiogenic mediators, like IL-1 and IL-6 mediate this process, which is reviewed elsewhere and may also be used diagnostically [69,70].”

We also included the important references suggested by the reviewer.

6) Unsupervised gene networking analysis help to identify new therapeutic target PMID: 32788346; 29042082), it can be discussed in 2.1.2.

We thank the reviewer for pointing out the relevance of unsupervised clustering. We have added refences to SC3 consensus clustering for single cell RNA sequencing as additional examples to t-SNE/PCA of unsupervised learning (see lines 278-281).

We also included the important references suggested by the reviewer (lines 298-300).

7) Single-cell RNA-sequencing is the latest approach to dissect the TME. Any implication of machine learning can be involved? Can be discussed in 2.3.2

We have added two paragraphs on single cell RNA sequencing for gastroesophageal cancers. “For instance, visualization techniques using principal component analysis (PCA) and t-statistic Stochastic Neighbor Embedding (t-SNE) are widely applied in the biomedical field, especially with a growing interest in single cell RNA/DNA sequencing [105-108].” “Owen et. al. harvested mucosa tissue from different anatomical locations of the stomach to identify an overlap between Barrett mucosa and found an association to submucosal glands by single-cell-RNA sequencing [144]. Here, and within other studies applying single-cell-RNA sequencing, SC3 consensus clustering as unsupervised learning method allowed to identify certain genes that would distinguish commonly in altered mucosa tissue [144,145].”

Reviewer 2 Report

Klein et al provided a brief review of GEA, its treatments, as well as understanding in its TME. The authors then provided a comprehensive review on the machine learning approaches that were adopted in understanding GEA and its TME. The review is very well-written and should provide immediate benefits to large audiences. 

Author Response

We thank the reviewer for his/her evaluation and positive remarks on our review.

Reviewer 3 Report

Comments to the authors

The review with the title “Machine Learning for future subtyping of the tumor microenvironment of gastro-esophageal adenocarcinomas” is in generally well done, but I would offer these comments to the authors: 

  • Some minor grammatical errors occur. The manuscript contains significant language-related issues. Please correct these types of grammatical errors throughout the paper.
  • Line 32- 33: “…pro-inflammatory cytokines recruiting several cell types…” Please add the major pro-inflamatory cytokines.
  • Line 122-125. The authors should mention the main cytokines which take parts in the formation of TME.
  • Line 127. “With more men than women being diseased of GEA...” please provide the exact percentage of men and women patients.
  • Line 133-134. It is the second time that authors mentioned the classification of GEA. I strongly recommend to provide the latest classification of GEA.
  • Line 141-151. Authors provide us with an interesting and comprehensive paragraph about the biomarkers that are linked to ICI response. I would suggest to add also the Microsatellite instability status (through the neo-antigens) as an additional marker for ICI response.
  • Please provide the figure legends.

Author Response

Point by point response to reviewers’ comments (in italics)

The review with the title “Machine Learning for future subtyping of the tumor microenvironment of gastro-esophageal adenocarcinomas” is in generally well done, but I would offer these comments to the authors: 

Some minor grammatical errors occur. The manuscript contains significant language-related issues. Please correct these types of grammatical errors throughout the paper.

We thank the reviewer for his/her comments on our work. We have substantially edited the manuscript.

Line 32- 33: “…pro-inflammatory cytokines recruiting several cell types…” Please add the major pro-inflamatory cytokines.

We have added examples of major pro-inflammatory cytokines: “For instance, increased expression of pro-inflammatory cytokines such as interleukin (IL)-1, IL-6, interferon (IFN)-, and tumor necrosis factor (TNF)-, leads to recruitment and activation of several cell types of immune and stromal cells that promote adaption of residual cells [4-6]” (lines 33-36)

Line 122-125. The authors should mention the main cytokines which take parts in the formation of TME.

We have added the main cytokines shaping the TME and added relevant references for a given reader to find information on this topic. “For instance, the pro-fibrotic and immunosuppressive transforming growth factor (TGF)-, or the pleiotropic immune mediators IL-1 and IL-6, mediate this process, which is also re-viewed elsewhere [5,6,72-75]. (lines 130-132)

Line 127. “With more men than women being diseased of GEA...” please provide the exact percentage of men and women patients.

We have added precise information of the latest global cancer statistics on the sex differences and provided age-standardized incidence rates for esophageal and stomach cancer: With more men than women being diseased of GEA (6.6 [8.2] / 1.8 [3.8]; men / women, region-specific incidence for age-standardized rates by sex for esophageal cancer [stomach] in 2020, Western Europe [21]). (lines 133-135)

Line 133-134. It is the second time that authors mentioned the classification of GEA. I strongly recommend to provide the latest classification of GEA.

We have added information on the TCGA gastric cancer classification: “As precision oncology emerges, classification of GEA such as the TCGA – chromosomal instable subtype [50%], microsatellite instable (MSI) high subtype [22%], genomically stable subtype [20%], Epstein–Barr virus-positive subtype [9%] among other classifications – are increasingly being used for prognostication. Unfortunately, these classifications have not yet been fully translated into improved therapeutic regimens, although MSI high subtypes and Epstein-Barr virus-positive cases show increased rates of response to ICI [82-87] (lines 141-147)

Line 141-151. Authors provide us with an interesting and comprehensive paragraph about the biomarkers that are linked to ICI response. I would suggest to add also the Microsatellite instability status (through the neo-antigens) as an additional marker for ICI response.

We thank the reviewer for his/her comment on our manuscript. “Moreover, integrative diagnostic approaches that combine several omics techniques have been shown to increase prediction to response to ICI therapy, including tumor-mutational burden (TMB) or neoantigen burden, to identify tumors with pro-immunogenic properties [103-105,108-112]. ” (lines 159-163)

“Unfortunately, these classifications have not yet been fully translated into improved therapeutic regimens, although MSI high subtypes and Epstein-Barr virus-positive cases show increased rates of response to ICI [82-87].” (lines 144-147)

Please provide the figure legends.

Thank you for pointing this out, it seems the figure legends were pasted in the main text instead of being associated to the figures itself. We have corrected the mistake.